# Effects of Community Water Fluoridation on Dental Caries Disparities in Adolescents

**DOI:** 10.3390/ijerph17062020

**Published:** 2020-03-19

**Authors:** Go Matsuo, Jun Aida, Ken Osaka, Richard Gary Rozier

**Affiliations:** 1Department of International and Community Oral Health, Tohoku University Graduate School of Dentistry, Sendai 9800872, Miyagi, Japan; aidajun@m.tohoku.ac.jp (J.A.); osaka@m.tohoku.ac.jp (K.O.); 2Department of Health Policy and Management, Gillings School of Global Public Health, University of North Carolina at Chapel Hill, Chapel Hill, NC 27599-7400, USA; gary_rozier@unc.edu

**Keywords:** oral health, dental caries, prevention, health disparities, community water fluoridation

## Abstract

Despite improvements in the prevalence of dental caries, disparities are still observed globally and in the U.S. This study examined whether community water fluoridation (CWF) reduced dental caries disparities in permanent teeth of 10- to 19-year-old schoolchildren in North Carolina. We used cross-sectional data representing K-12 schoolchildren in North Carolina (NC) public schools. A poisson regression model was used to determine whether the association between children’s parental educational attainment and the prevalence of dental caries of children differed by children’s lifetime CWF exposure. We analyzed data on 2075 students. Among the children without any CWF exposure in their life, statistically significant caries disparities by parental educational attainment were observed. Compared to the children of parents with more than high school education, the relative risk for those with a parent with a high school education was 1.16 (95% CI = 1.01, 1.33) and those with less than a high school education was 1.27 (95% CI = 1.02, 1.60). In contrast, these disparities were not observed among children exposed to CWF throughout their lives. Socioeconomic disparities in dental caries were not observed among 10–19-year-old schoolchildren with lifetime CWF exposure. CWF seemed to reduce dental caries disparities.

## 1. Introduction

Despite a decline in the prevalence of dental caries, untreated caries in permanent teeth was the most prevalent condition found in the Global Burden of Disease Study [1]. This issue is also is characterized by persistent global inequalities [2,3,4]. The high prevalence of dental caries and its inequalities also are still observed in the U.S. [5]. Eliminating oral health disparities is among the national goals of the U.S. [6,7].

Interventions to address health disparities need to be carried out at multiple levels using upstream approaches [8,9,10]. Policies for structural changes in the environment, such as community water fluoridation (CWF) or healthy food and nutrition, are important upstream strategies in addition to usual policies focusing on proximal influences on oral health such as oral health behaviors [10,11]. 

The benefits of water fluoridation on dental caries reductions are well documented, but its effects on oral health inequalities, although reported globally, are not as well studied. A systematic review published in 2000 concluded that there was a reduction in disparities in dental caries in primary and permanent teeth across social classes in five and 12-year-olds due to CWF when counts of affected teeth were used as the outcome. The review also found insufficient evidence of any impact on disparities when the proportion of caries-free children was used as the outcome or among the other age groups [12]. Following the “York Review”, a few other studies investigated the effect of CWF on oral health inequalities in different countries [13,14,15,16,17]. Only one of these studies was conducted in the U.S. [18]. Although the complete elimination of caries disparities seems difficult [16], all these studies showed a trend toward a reduction in disparities through the use of CWF. 

However, the previous studies have some limitations. One was the measurement of exposure to CWF. The CWF status was classified according to whether the water supply in the study population’s area of residence or its school water system was fluoridated at the time of data collection. The U.S study determined CWF exposure according to the proportion of the population in the county of residence at the time of data collection (<75% of the population, vs. ≥75%) [18] served by a fluoridated community water system. People often change where they live throughout their lives, which in turn can change their exposure to CWF. The cross-sectional aspect of the data limits the accuracy of the effect of CWF on oral health disparities. 

While most studies used the latest income status, some used race as the measure of socioeconomic status (SES). Only a few used educational attainment of an adult family member, and no study in the U.S. has analyzed the effect of CWF on oral health disparities according to the education level of the child’s parent. 

Studies on the effects of CWF on dental caries have used diagnostic criteria that measure obvious disease, done in order to achieve an acceptable degree of examination reliability. But this approach excludes some subjects with the disease in its earliest stages. In this study, we include lesions in the initial stages of development in our assessments, which should provide new insights into the effects of CWF on dental caries disparities.

The purpose of this study is: (1) To examine the effects of CWF on dental caries prevalence, and (2) to determine its effects on disparities in dental caries prevalence according to educational attainment.

## 2. Methods 

### 2.1. Study Design and Participants

This study used data collected as part of the 2003–2004 Statewide Dental Survey of North Carolina School Children (SNCSC), conducted by the Oral Health Section (OHS) of the North Carolina Division of Public Health. This survey collected clinical and parental-reported data from a probability sample of all students in grades K through 12 in North Carolina and examined 7686 students. Studies have used the information from this survey to evaluate the effects of fluoride mouth rinse programs on dental caries, the effects of enamel fluorosis and dental caries on oral health-related quality of life, and statistical methods applied to the study of oral health conditions [19,20,21]. 

The sampling strategy for the SNCSC was a stratified cluster design in which the sampling units consisted of classrooms identified by the teacher, grade level, and subject. The classroom in the elementary school strata was stratified according to the degree of urbanism for the country (urban vs. rural), caries risk status of the school (high vs. other), which was determined by the proportion of children receiving free or reduced lunch, and the existence of a school-based fluoride mouth rinse program (yes vs. no). The classrooms in the middle and high school grades were stratified by the degree of urbanism of the county and the percentage of the Latino population. All the students from selected classrooms were eligible for participation in the survey. 

Ten trained and calibrated dentists conducted a clinical survey. Prior to the clinical examination, the teeth were cleaned with a dry toothbrush and dried using compressed air. The Kappa values for the caries status of each surface (decayed, missing or filled) was 0.79, and the intraclass correlation coefficient for the child-level count of total carious surfaces was 0.88 [20]. For purposes of this analysis, older children, 10 through 19 years of age, with available clinical and questionnaire data were included to provide a sufficient amount of time for children to develop dental caries.

### 2.2. Response Variable

This survey categorized carious lesions according to the World Health Organization (WHO) as non-cavitated lesions (also referred to as pre-cavitated or white spot lesions, or D_1_ Lesions) and cavitated lesions (frank cavitation, either in enamel only or into dentin, or D_2–3_ lesions). For this research, we combined D_1_ and D_2–3_ into D_1–3_. Our response variable was developed from the D_1-3_MFS score, a count of the surfaces of teeth with dental caries, missing surfaces because of the extraction of teeth with dental caries, and filled surfaces (treated dental caries). For the analysis, the D_1–3_MFS index score (DMFS) for each examined student was dichotomized as no caries surfaces (DMFS = 0) versus some caries surfaces (DMFS ≧ 1). 

### 2.3. Primary Explanatory Variable

Two primary explanatory variables—CWF exposure history, and the parents’ educational attainment—were used in the study.

For each child, parents reported in the written questionnaire on the current main home drinking water source (“Home well or spring”, “Bottled water”, or “City or community water system”). If they had lived elsewhere, their previous address and past primary drinking water source were collected. The name of the city/community water supply was also reported and used to determine the fluoridation status of water at that time the family was living there. Based on that information, we categorized CWF exposure for each year of the child’s life as optimal CWF (0.7 to 1.2 ppm) or not. Then, we calculated the proportion of the child’s lifetime with optimal CWF exposure by dividing the CWF exposed years by their age in years. The detail process for measuring CWF exposure levels is available in the Appendix A. 

We assessed socioeconomic disparities in dental caries using parental educational attainment. According to a systematic review on socioeconomic inequality and dental caries, the measures of SES generally used for children’s oral health studies are parental educational, income, and occupational background [3]. Because parental educational background could determine their current and past income status, we believe that educational attainment relates to the past and current SES status and is as important an SES measure as the income level. No such study has been conducted in the U.S. using parental education. From the parental questionnaire, parents’ educational attainment was noted with the question “What is the highest grade in school that you have completed?”. The options “Up to 8^th^ grade”, “Some high school”, “High school graduate”, “Some college”, “College graduate or some professional school”, “Don’t want to answer”. For this study, we developed categories excluding “Don’t want to answer”, which were “More than high school graduate”, “high school graduate” and “Less than high school graduate”. The “More than high school graduate” included “Some college”, “College graduate or some professional school”, and the “Less than high school graduate” included “Up to 8^th^ grade”, “Some high school”.

### 2.4. Control Variables

Demographic information from parent questionnaires consisted of age, sex (male or female), race (White, Black, Hispanic), parental marital status, insurance status, and history of professional fluoride application. The clinical data provide a covariate for the placement of dental sealants. 

### 2.5. Statistical Analyses

First, we developed a descriptive table for all students with information on CWF exposure (*N* = 2075) and for students stratified by exposure to CWF throughout their entire life (100% exposed group, *N* = 1057) or were never exposed (0%, *N* = 815). Percentage distributions were compared using ANOVA for variables with more than two groups and Chi-Square tests for variables with two groups. 

We chose to compare only the two extreme exposure groups (0% exposed and 100% exposed) because the magnitude of CWF effects from partial lifetime exposures will depend heavily on the age of exposures. For example, if children were exposed to CWF for only half their life, whether the first half or the latter half, the exposure would result in different preventive effects on dental caries in permanent teeth because of a number of factors including eruption patterns, changes in diet and other environmental factors. We decided to control for the potential of exposure age on outcomes through limiting the sample to the two exposure groups rather than through analysis considerations.

Secondly, we ran Poisson regression models to determine the variables that had a strong influence on caries prevalence. We chose Poisson regression using robust variance estimations because the prevalence of the disease was more than 10% in our data [22]. First, we included sex, age and parental education level (Model 1); then, we added the remaining variables (Model 2) using the data from all the students. Next, we developed Model 3 including the two fluoridation exposure groups—those exposed to CWF throughout their lives and those who were never exposed—to assess the association of education level in each of the exposure groups while controlling for potential confounders. 

Finally, we calculated population average model-adjusted risks at each parental educational level and calculated the average marginal effects between the highest education level compared with each of the other levels and whether the risk differences were statistically significant.

We used Stata version 12 (StataCorp LP, College Station, TX, USA). To account for the sample design, we performed all analyses with normalized survey weights and a classroom cluster definition, producing estimates for the NC school population. For all models, we report prevalence ratios with corresponding 95% confident intervals. We performed all tests using a 0.05 significant level. This study was approved by the Ethics Committee of the Tohoku University Graduate School of Dentistry. IRB approval for the parent study was provided by the University of North Carolina at Chapel Hill, North Carolina, U.S.A.

## 3. Results

The analytical sample consisted of 2075 students aged 10–19 years. Descriptive statistics are presented in Table 1. The percentage of students with one or more DMFS differed according to parents’ educational attainment, with a smaller prevalence in students having highly educated parents. The percentage of students with at least one DMFS was 59.2% in those whose parents had more than a high school degree and 70.32% in those whose parents had only completed high school. A similar trend was observed among students without any CWF exposure (0% group) and those with full exposure (100% group). 

The results of the multivariate Poisson regression analysis are presented in Table 2. In Model 1, students whose parents were less educated had a statistically significant greater dental caries experience than those whose parents were highly educated (high school graduate: Prevalence ratio (PR) = 1.18; 95% confidence interval (CI) = 1.08, 1.28; less than high school graduate: PR = 1.16; 95% CI = 1.01, 1.32). This trend also was observed in Model 2 (high school graduate: PR = 1.16; 95% CI = 1.06, 1.28; less than high school: PR = 1.09; 95% CI = 0.90, 1.32). 

Some other variables were significant in the regression models. Compared to white children, black children had a higher prevalence of dental caries (black: PR = 1.24; 95% CI = 1.10, 1.39; hispanics: PR = 1.09; 95% CI = 0.86, 1.38). Compared to the 0% fluoridation exposure group, the 100% exposure group had significantly lower caries prevalence (PR = 0.86; 95% CI = 0.77, 0.96).

Results of the multivariate Poisson regression stratified by fluoridation exposure are displayed in Table 3. In the 0% exposure group, dental caries disparities based on parents’ education level were observed. Compared to the most highly educated group, the high school graduate and less than high school completion groups had statistically significantly higher dental caries prevalence (high school graduate: PR = 1.16; 95% CI = 1.01, 1.33; less than high school graduate: PR = 1.27; 95% CI = 1.02, 1.60). In the 100% fluoridation exposure group, there were no differences in dental caries prevalence based on parents’ education level. 

Figure 1 shows the adjusted population average probabilities of having any DMFS according to the educational status of parents and fluoridation exposure. Among the 0% exposure group, caries experience was significantly higher for students with lesser-educated parents than in those with the highest educated parents (difference among high school graduate = 10% points higher; *p* < 0.05, less than high school graduate = 17% points higher; *p* < 0.05). No statistically significant difference was observed in the education level among the 100% fluoridation exposure group.

Average marginal effects between “High school graduate” and “Less than high school graduate”, and “More than high school graduate” were calculated for each community water fluoridation exposure level. (* *p* < 0.05).

## 4. Discussion

Like many others, this study revealed the association of CWF with the prevention of dental caries. The 100% exposure group had a lower prevalence of dental caries than the 0% exposure group. Further, parents’ educational level was associated with prevalence as well, indicating the existence of socioeconomic disparities in dental caries. Importantly, this association was attenuated by community water fluoridation. The 0% fluoride exposure group was observed to have significant differences in dental caries prevalence according to parents’ educational status, while no differences were found in the 100% exposure group. 

One of the strengths of the present study is that we used a retrospective history of CWF exposure for each year of life rather than the information from one point in time. Previous studies have used information on the CWF status of the area where the children were living at the time of the survey or their school’s water system for the measurement of exposures [13,14,15,16,17,18]. A novel aspect of our study is that we accounted for the duration of exposures. These data allowed us to explore with more precision the association of exposures with the outcomes. We also used more precise clinical data compared to those used in previous studies. Most previous studies have used D_2-3_MFT or D_2-3_MFS data rather than D_1-3_MFS. Measurement of incipient lesions, or dental caries in its earliest stages, enabled us to have a more sensitive measure of the effects of CWF on dental caries disparities. Finally, unlike many previous studies, we chose parental education as our measure of socioeconomic status [13,15,16,18]. We believe that education status is equally as important as income status in disparities studies [3]. Thus, because of the nature of our studies’ strengths, we believe our findings add valuable information to the existing literature [13,14,15,16,17,18]. 

One of the limitations of this study is recall bias in reporting residency histories and primary drinking water source, and thus fluoride exposures. The parents of those children who have good oral health without CWF exposure might also have good oral health literacy. Some families, despite choosing to drink bottled water, might report a city water system as the main source of drinking water because they are aware of the effects of CWF. Such bias may have limited our ability to explore more precise mechanisms of the effect of health differences. 

A second limitation is the study’s cross-sectional design. The associations observed between explanatory variables and oral health outcomes may not be causal. We might have excluded some important determinants of the oral health status of children, such as access to dental care, parent’s health literacy, and children’s oral health behaviors and practices that could have confounded our findings [23]. 

We found that only about a half of children had been exposed to optimal levels of CWF for their entire life. Between 53% to 58% of the total NC population was drinking fluoridated water between 1985 to 2002, roughly the relevant years for this study sample. Estimates from other studies that could be used for comparison generally are not available. This study thus provides important and practical information to help guide efforts to ensure maximum fluoride exposure of the population in a large geographic area. Some children who were placed in the 100% CWF exposure group used bottled water to some extent. This may have led to an underestimation of the effect of CWF in the current study. 

In the present study, the reduction of dental caries attributed to water fluoridation and differences according to SES groups was observed. These results are mostly consistent with those of previous studies performed in various countries, including the U.S. [13,14,15,16,17,18]. Cho et al. [13] and Kim et al. [14] showed that socioeconomic inequality exists in non-CWF areas but not in CWF areas in Korea. Cho et al. used the Family Affluence Scale (FAS), and Kim et al. used education, income, and FAS as the measures of SES status. Cho et al. used DMFT (a count of the teeth with dental caries, missing because of the extraction due to dental caries, and treated dental caries), and Kim et al. used DMFT, DMFS, and caries prevalence as measures of dental caries experience. Do et al. [15] conducted their study in Australia and explored dental health disparities based on income and race. They concluded that inequalities in caries progression in the primary and permanent dentitions were usually greater in non-CWF areas than in CWF areas. Sanders et al. [18] reported findings from the U.S. using national data. They concluded that reductions in absolute and relative fluoridation-related caries experience were most pronounced for the lowest income group. However, for permanent teeth, the attenuation of an income gradient in DMFS was smaller than that for primary teeth. 

Our study showed no effect for CWF on racial disparities. Only a few studies have found an effect of CWF on these disparities, and no study in the U.S. has been conducted to date. The causes of racial and ethnic disparities have not been fully explained in previous research. Further studies are needed to explore the causal pathways of these disparities and the effects that current preventive interventions have on these pathways.

Two previous studies in New Zealand and Canada did not provide positive conclusions on the effects of water fluoridation on dental disease disparities [16,17]. Schluter & Lee [16] concluded that CWF did not eliminate the differences between indigenous and non-indigenous race groups. However, although differences might not have been completely eliminated, the trend of reduction was observed. McLaren & Emercy [17] provided mixed conclusions and reported that the effect of CWF was especially pronounced in the lower education group and higher income group compared to their counterparts. Usually, the higher income group has a lower health risk. However, the highest income group among non-CWF exposed areas had the worst oral health condition in their study. Therefore, greater benefit was observed in the group with poor oral health status as the other studies reported.

Based on the results of the present and previous studies, the benefit of CWF is likely to be larger among people with lower SES who are often at high risk for dental caries. Therefore, CWF works as the proportionate universalism approach in which it is crucial to reduce the steepness of the social gradient in health. Unfortunately, some population strategies could increase health disparities along with a reduction in the mean disease incidence [24]. Therefore, the concept of “proportionate universalism” which is a population approach with a better effect on socially compromised, high-risk population than other groups has been proposed for use in the context of health inequalities [25]. There is no single method to solve the dental caries problem. As proposed by WHO, multilevel, upstream interventions are needed to address this public health problem [26]. CWF is also cost-effective [27,28]. Hence, it should be chosen as a primary intervention strategy to tackle disparities in dental caries.

## 5. Conclusions

We found that CWF reduced dental caries and its disparities. The results of this study support the inclusion of CWF as a primary public health intervention to reduce dental caries and oral health disparities.

## Figures and Tables

**Figure 1 ijerph-17-02020-f001:**
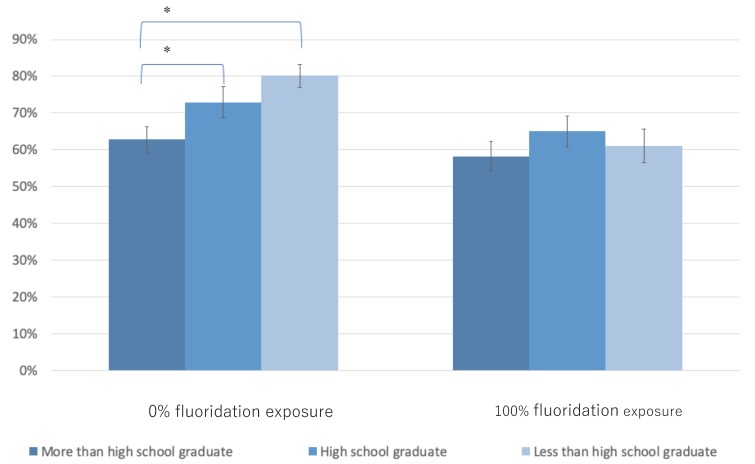
Adjusted population average probability of caries experience in children by fluoridation exposure level and parental educational attainment.

**Table 1 ijerph-17-02020-t001:** Basic characteristics of the study participants.

		All (*N* = 2075)				Fluoridation 0% (*N* = 815)				Fluoridation 100% (*N* = 1057)			
		*n*	Percent	Proportion D_1–3_MFS > 0	*p*	*n*	Percent	Proportion D_1–3_MFS>0	*p*	*n*	Percent	Proportion D_1–3_MFS > 0	*p*
Sex (*N* = 1858)						*N* = 733				*N* = 945			
	Male	866	46.61	62.59		365	49.80	63.56		421	44.55	62.71	
	Female	992	53.93	64.21		368	50.20	69.57		524	55.45	60.88	
Age (*N* =1858)						*N* = 733				*N* = 945			
	10	313	16.85	53.35		119	16.23	55.46		162	17.14	54.32	
	11	269	14.48	52.04		111	15.14	56.76		125	13.23	50.4	
	12	284	15.29	52.82		118	16.10	60.17		140	14.81	47.86	
	13	199	10.71	67.34		84	11.46	72.62		100	10.58	60	
	14	192	10.33	69.27		75	10.23	73.33		101	10.69	67.33	
	15	184	9.9	75		85	11.60	70.59		85	8.99	78.82	
	16	171	9.2	76.02		63	8.59	76.19		85	8.99	72.94	
	17	169	9.1	73.96		59	8.05	84.75		94	9.95	68.09	
	18	71	3.82	80.28		16	2.18	75.00		50	5.29	82	
	19	6	0.32	83.33	**	3	0.41	66.67	**	3	0.32	100	**
Parental educational attainment (*N* = 1736)						*N* = 692				*N* = 873			
	More than high school graduate	1094	63.02	59.23		397	57.37	62.22		573	65.64	57.77	
	High school graduate	475	27.36	70.32		221	31.94	71.04		212	24.28	68.87	
	Less than high school graduate	167	9.62	70.06	*	74	10.69	74.32	**	88	10.08	68.18	**
Race (*N* = 1762)						*N* = 698				*N* = 891			
	White	1175	66.69	59.23		527	75.50	63.95		506	56.79	54.74	
	Black	486	27.58	72.02		138	19.77	76.81		326	36.59	70.55	
	Hispanic	101	5.73	65.35	**	33	4.73	54.55		59	6.62	71.19	**
Parental marital status (*N* = 1738)						*N* = 689				*N* = 876			
	Single	482	27.73	65.56		161	23.37	70.19		283	32.31	64.31	
	With partner	1256	72.27	62.26		528	76.63	64.39		593	67.69	60.54	
Insurance status (*N* = 1687)						*N* = 669				*N* = 852			
	Commercial insurance	774	45.88	58.40		290	43.35	61.03		393	46.13	56.49	
	Public insurance	398	23.59	70.85		164	24.51	74.39		203	23.83	69.46	
	No insurance	515	30.53	63.50	*	215	32.14	66.51		256	30.05	60.55	
Professional fluoride application (*N* = 1567)						*N* = 628				*N* = 779			
	No experience	364	23.23	62.91		156	24.84	64.74		185	23.75	62.7	
	At least once	1203	76.77	61.51		472	75.16	66.31		594	76.25	57.58	
Dental sealant status (*N* = 1858)						*N* = 733				*N* = 945			
	No	781	42.03	63.64		315	42.97	65.08		403	42.65	65.26	
	At least one	1077	57.97	63.32		418	57.03	67.70		542	57.35	59.04	
Fluoridation exposure level (*N* = 1858)													
	0%	733	39.45	66.58									
	0%<, ≤50%	103	5.54	57.28									
	50%<, <100%	77	4.14	63.64									
	100%	945	50.86	61.69									

Differences among each group were examined using ANOVA for comparing more than two groups, and chi-square test for comparing two groups (** *p* < 0.01, * *p* < 0.05).

**Table 2 ijerph-17-02020-t002:** Results of Poisson models for all samples.

	Prevalence Ratios (95% Confidence Interval)	Model 1 (*N* = 1735)	Model 2 (*N* = 1272)
Sex			
	Male (Ref)		
	Female	1.01 (0.93, 1.09)	1.04 (0.95, 1.14)
Age			
		1.06 ** (1.04, 1.08)	1.07 ** (1.05, 1.09)
Parental educational attainment			
	More than high school graduate (Ref)		
	High school graduate	1.18 ** (1.08, 1.28)	1.16 **(1.06, 1.28)
	Less than high school graduate	1.16 * (1.01, 1.32)	1.09 (0.90, 1.32)
Race			
	White (Ref)		
	Black		1.24 ** (1.10, 1.39)
	Hispanic		1.09 (0.86, 1.38)
Parental marital status			
	Single (Ref)		
	With partner		0.99(0.89, 1.11)
Insurance status			
	Commercial insurance (Ref)		
	Public insurance		1.11 (0.98, 1.26)
	No insurance		1.06 (0.95, 1.18)
Fluoridation exposure level			
	0% (Ref)		
	0%<, ≤50%		1.01 (0.82, 1.24)
	50%<, <100%		0.99 (0.81, 1.22)
	100%		0.86 ** (0.77, 0.96)
Professional fluoride application			
	No experience (Ref)		
	At least once		1.05 (0.92, 1.2)
Dental sealant status			
	No (Ref)		
	At least one		1.09 (0.98, 1.2)
Cons			
		0.28 ** (0.22, 0.36)	0.20 ** (0.14, 0.29)

(** *p* < 0.01, * *p* < 0.05).

**Table 3 ijerph-17-02020-t003:** Results of Poisson models for fluoridation 0% exposed group and 100% exposed group.

	Prevalence Ratios (95% Confidence Interval)	Fluoridation 0% Exposed (*N* = 520)	Fluoridation 100% Exposed (*N* = 615)
Sex			
	Male (Ref)		
	Female	1.02 (0.90, 1.18)	1.07 (0.92, 1.24)
Age			
		1.06 ** (1.02, 1.09)	1.08 ** (1.05, 1.11)
Parental educational attainment			
	More than high school graduate (Ref)		
	High school graduate	1.16 * (1.01, 1.33)	1.11 (0.95, 1.30)
	Less than high school graduate	1.27 * (1.02, 1.60)	1.05 (0.76, 1.45)
Race			
	White (Ref)		
	Black	1.24 ** (1.06, 1.45)	1.25 ** (1.06, 1.48)
	Hispanic	0.90 (0.63, 1.27)	1.14 (0.75, 1.74)
Parental marital status			
	Single (Ref)		
	With partner	0.98 (0.84, 1.15)	1.06 (0.90, 1.23)
Insurance status			
	Commercial insurance (Ref)		
	Public insurance	1.08 (0.92, 1.26)	1.19 (0.99, 1.42)
	No insurance	1.04 (0.87, 1.23)	1.06 (0.90, 1.24)
Professional fluoride application			
	No experience (Ref)		
	At least once	1.11 (0.90, 1.35)	1.02 (0.86, 1.22)
Dental sealant status			
	No (Ref)		
	At least one	1.09 (0.93, 1.28)	1.02 (0.89, 1.18)
Cons			
		0.24 ** (0.14, 0.41)	0.15 ** (0.09, 0.25)

(** *p* < 0.01, * *p* < 0.05).

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
