# Peer review of "Effects of Community Water Fluoridation on Dental Caries Disparities in Adolescents"

_ijerph, 2020, doi:10.3390/ijerph17062020_

Round 1
Reviewer 1 Report
Manuscript ijerph-722705 aimed to examine effect of CWF on dental caries and education-related inequalities in dental caries. This is an important topic.
The study was well conducted. Below are some comments to further improve the manuscript.
- The authors conducted stratified analysis by exposure to CWF. This is a good method. However, I would ask if the authors also consider testing interaction between education and exposure to CWF.
- The independent variables included both child-level and parent-level. It should make it clearer in the tables and the text. For example, Education and Marital status were parental factors.
- There are some discrepancies in number of children by CFW groups. At the top row in table 1, there were 815 and 1057 children in the 0% and 100% groups. The first column at the end of the table shows that there were 733 and 945 children in those exposure groups. Please explain.
- The study was conducted in NC. While there was no US IRB?
Author Response
Thank you very much for your feedback. Here are our responses.
- We considered testing the interaction between education and exposure in our analysis. However, we decided to go with the stratification analysis because, as mentioned in the text, it is difficult to interpret the preventive effects of CWF in the middle groups of CWF exposure levels (0%<, <=50% & 50%<, <100%). For example, if children were exposed to CWF for only half their lives (50%), whether the first half or the latter half, the exposure could result in different preventive effects on dental caries in permanent teeth because of several factors including eruption patterns. In addition, we have extremely small samples in each education levels among middle groups of CWF exposure. Therefore, we believe that stratification is a better way to approach the analysis for our research question and data.
- In the text, we add “parental” before marital status. Also, in the tables, we changed “education” to “parental educational attainment”, and “marital status” to “parental marital status”.
- The numbers at the top of the table 1 show the number of the children who had non-missing information for CWF exposure levels for each group. However, the last rows of the first column show the number of each group of the fluoridation exposure with clinical data.
If we include only children who have the information of CWF
- 0% exposure: 815 (39.28%)
- 0<, <=50%: 115 (5.54%)
- 50%>, <100%: 88 (4.24%)
- 100%: 1,057 (50.94%)
- Total: 2,075 (100%)
If we include children who have both the information of CWF and clinical data.
- 0% exposure: 733 (39.45%)
- 0<, <=50%: 103 (5.54%)
- 50%>, <100%: 77 (4.14%)
- 100%: 945 (50.86%)
- Total: 1,858 (100%)
We added the following information in the text: “First, we developed a descriptive table for all students with information on CWF exposure (N=2,075) and for students stratified by exposure to CWF throughout their entire life (100% exposed group, N=1,057) or never exposed (0%, N=815).”
- IRB approval for the parent study was provided by the University of North Carolina at Chapel Hill IRB # 03-2062. I added the information in the text.
Postscript: As we prepared the response, I double checked the numbers of all tables and figure, and noticed that some numbers need minor changes. None of those corrections provide any changes to our conclusion, except our comment on the percentage of the children who were exposed to CWF throughout their entire life. We made the necessary correction to the text related to this (line 287-290).
Reviewer 2 Report
The authors are using cross-sectional data from 2003-2004 to assess the role of lifetime exposure to community water fluoridation on caries in permanent teeth in children ages 10-19 with a particular focus on socioeconomic disparities. This is an important topic.
The key findings are compelling, in that, parental education level was found to be more important for describing risk among children living in non-fluoridated communities. This suggests that Community Water Fluoridation (CWF) is effective in addressing socioeconomic disparities.
Major concerns
- Racial disparities are not affected by CWF in this analysis. Please discuss this. Is systemic racism impacting access to care in North Carolina?
- I am confused about the sample sizes in the sizes for the analysis. In column 1 of table 1 it looks like CWF=0 is 733 and CWF=100 is 945 but that does not fit with the numbers in Table 3 or columns 2 and 3 of Table 1. Please clarify.
Minor concern
3. In the Abstract, please revise 1,16 to read 1.16 in line 20.
Author Response
Thank you very much for your feedback. Here are our responses.
- According to the literature review, there are only a few non-U.S. studies that have investigated the effects of CWF on racial disparities, and no studies in the U.S. This suggests a lack of understanding of the association of CWF on racial disparities under the current preventive environment. Also, the causes of racial/ethnic disparities have not been fully explained in previous research. Further studies are needed to explore causal pathways of these potential disparities and their solution. We added similar content in the discussion.
- The numbers at the top of the table 1 show the number of the children who had non-missing information for CWF exposure levels for each group. However, the last rows of the first column show the number of each group of the fluoridation exposure with clinical data.
If we include only children who have the information of CWF
- 0% exposure: 815 (39.28%)
- 0<, <=50%: 115 (5.54%)
- 50%>, <100%: 88 (4.24%)
- 100%: 1,057 (50.94%)
- Total: 2,075 (100%)
If we include children who have both the information of CWF and clinical data.
- 0% exposure: 733 (39.45%)
- 0<, <=50%: 103 (5.54%)
- 50%>, <100%: 77 (4.14%)
- 100%: 945 (50.86%)
- Total: 1,858 (100%)
We added the following information in the text: “First, we developed a descriptive table for all students with information on CWF exposure (N=2,075) and for students stratified by exposure to CWF throughout their entire life (100% exposed group, N=1,057) or never exposed (0%, N=815).” In the first version of the manuscript I put the number of the children with CWF exposure information. In the current version, changed them to the number of the children included in the analysis in table 2 and table 3. Some observations were lost because of missing data on one or more variables.
- Thank you. We revised it.
Postscript: As we prepared the response, I double checked the numbers of all tables and figure, and noticed that some numbers need minor changes. None of those corrections provide any changes to our conclusion, except our comment on the percentage of the children who were exposed to CWF throughout their entire life. We made the necessary correction to the text related to this (line 287-290).
Round 2
Reviewer 2 Report
The authors have addressed my concerns.